

**Patterns of nitrogen and phosphorus pools in terrestrial ecosystems in China**
Yi-Wei Zhang[1#], Yanpei Guo[1#], Zhiyao Tang[1]*, Yuhao Feng[1], Xinrong Zhu[1], Wenting Xu[2],
Yongfei Bai[2], Guoyi Zhou[3], Zongqiang Xie[2], Jingyun Fang[1]
[1]Institute of Ecology, College of Urban and Environmental Sciences and Key Laboratory for
Earth Surface Processes of the Ministry of Education, Peking University, Beijing 100871
[2]State Key Laboratory of Vegetation and Environmental Change, Institute of Botany, Chinese
Academy of Sciences, Beijing 100093
[3]Institute of Ecology, Jiangsu Key Laboratory of Agricultural Meteorology, Nanjing University
of Information Science & Technology, Nanjing 210044, China
[#]Equal contribution
Corresponding author:
Zhiyao Tang, Ph.D.
E-mail: zytang@urban.pku.edu.cn
Tel/Fax: +86-10-6275-4039



**Abstract**
Recent increases in atmospheric carbon dioxide ($CO_2$) and temperature relieve the limitation
of these two on terrestrial ecosystem productivity, while nutrient availability constrains the
increasing plant photosynthesis more intensively. Nitrogen (N) and phosphorus (P) are critical
for plant physiological activities and consequently regulates ecosystem productivity. Here, for
the first time, we mapped N and P densities of leaves, woody stems, roots, litter and soil in
forest, shrubland and grassland ecosystems across China, based on an intensive investigation
in 4175 sites, covering species composition, biomass, and nutrient concentrations of different
tissues of living plants, litter and soil. Forest, shrubland and grassland ecosystems in China
stored $7665.62 \times 10^6$ Mg N, with $7434.53 \times 10^6$ Mg (96.99%) fixed in soil (to a depth of one
metre), and $32.39 \times 10^6$ Mg (0.42%), $59.57 \times 10^6$ Mg (0.78%), $124.21 \times 10^6$ Mg (1.62%) and
$14.92 \times 10^6$ Mg (0.19%) in leaves, stems, roots and litter, respectively. The forest, shrubland
and grassland ecosystems in China stored $3852.66 \times 10^6$ Mg P, with $3821.64 \times 10^6$ Mg
(99.19%) fixed in soil (to a depth of one metre), and $3.36 \times 10^6$ Mg (0.09%), $14.06 \times 10^6$ Mg
(0.36%), $11.47 \times 10^6$ Mg (0.30%) and $2.14 \times 10^6$ Mg (0.06%) in leaves, stems, roots and
litter, respectively. Our estimation showed that N pools were low in northern China except
Changbai Mountains, Mount Tianshan and Mount Alta, while relatively higher values existed
in eastern Qinghai-Tibetan Plateau and Yunnan. P densities in plant organs were higher
towards the south and east part of China, while soil P density was higher towards the north
and west part of China. The estimated N and P density datasets, "Patterns of nitrogen and
phosphorus pools in terrestrial ecosystems in China" (the pre-publication sharing link:
https://datadryad.org/stash/share/78EBjhBqNoam2jOSoO1AXvbZtgIpCTi9eT-eGE7wyOk),
are available from the Dryad Digital Repository (Zhang et al., 2020). These patterns of N and



P densities could potentially improve existing earth system models and large-scale researches
on ecosystem nutrients.


**Key words:** climate; nitrogen pools; phosphorus pools; nutrient limitation; spatial distribution



## 1    Introduction


Nitrogen (N) and phosphorus (P) play fundamental roles in plant physiological activities
and functioning, such as photosynthesis, resource utilization and reproductive behaviours
(Fernández-Martínez et al., 2019; Lovelock et al., 2004; Raaimakers et al., 1995), ultimately
regulating plant growth and carbon (C) sequestration efficiency (Terrer et al., 2019). Under the
background of global warming, the limiting factors for the plant growth, such as carbon dioxide
($CO_2$) and temperature, are becoming less restrictive for terrestrial ecosystem productivity
(Norby et al., 2009), while nutrient availability tends to constrain the increasing plant
photosynthesis more intensively (Cleveland et al., 2013; Du et al., 2020). As the key nutrients
for plant growth, N and P independently or together limit biomass production (Elser et al., 2007;
Finzi et al., 2007). N influence $CO_2$ assimilation in various ways (Vitousek and Howarth, 1991).
For example, N is a critical element in chlorophyll (Field, 1983), and plant metabolic rates are
also regulated by N content (Elser et al., 2010). P is crucial in RNA and DNA construction, and
its content is associated with water uptake and transport (Carvajal et al., 1996; Cheeseman and
Lovelock, 2004) as well as energy transfer and exchange (Achat et al., 2009). P shortage could
lower photosynthetic C-assimilation rates (Lovelock et al., 2006).
In spite of the key importance of N and P for plants, knowledge on the patterns of their
storage in terrestrial ecosystems are limited. With additional $CO_2$ entering atmosphere, more N
could be allotted to plant growth and soil organic matter (SOM) accumulation, which may lead
to less available mineral N for plant uptake (Luo et al., 2004). Direct and indirect evidences
show that N limits productivity in temperate and boreal areas (Bonan, 1990; Miller, 1981;
Vitousek, 1982). P originates from bedrock weathering and litter decomposition in terrestrial
ecosystems, and it experiences long-term biogeochemical processes before available to plants



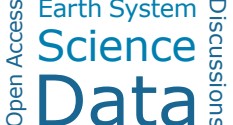

(Föllmi, 1996), which consequently makes P a more predominant limiting factor to ecosystem
productivity (Reed et al., 2015). Additionally, P decomposition rates are constrained by limited
soil labile P storage, especially in tropical forests where soil P limitation is extreme (Fisher et
al., 2012).
Ecosystem models based on Amazon forest free air $CO_2$ enrichment (FACE) experiments
consistently showed that biomass C positively responded to simulated elevated $CO_2$, but the
models incorporating N and P availability showed lower plant growth than those not (Wieder
et al., 2015). Moreover, a recent study suggested that the inclusion of N and P availability into
the earth system models (ESMs) remarkably improved the estimation accuracy of C cycles over
previous models (Fleischer et al., 2019). Hence, understanding and predicting the patterns and
mechanisms of global C dynamics require well characterizing of N and P conditions.
N and P pools in ecosystems consist of several components that cast different influences
on ecosystem C storages and fluxes. For example, N and P in plants directly affect C
sequestration (Thomas et al., 2010), but their activities differ among organs (Elser et al., 2003;
Parks et al., 2000); the soil pools are the source of plant nutrition; and the litter pools act as a
transit link that returns nutrients from plants to soil (McGrath et al., 2000). Thus, an accurate
estimation of ecosystem N and P pools involves calculating specific nutrient densities in all
these components.
Terrestrial ecosystems in China play a considerable part in the continental and global C
cycles. Satellite data verified that China contributed to a 1/4 of global net increase in leaf area
from 2000 to 2017 (Chen et al., 2019). The total C pool in terrestrial ecosystems in China is
79.2 Pg C, and this number is still growing because of the nationwide ecological restoration
constructions, which accounted for 56% of the total C sequestration in the restoration area in



China from 2001 to 2010 (Lu et al., 2018). N and P limitations are ubiquitous in natural
ecosystems in China (Du et al., 2020).    Understanding the distribution and allocation of N and
P in ecosystems is of great significance for a precise projection of C cycle in China. Although
there are a few studies on the spatial patterns of soil nutrient storages in China (Shangguan et
al., 2013; Yang et al., 2007; Zhang et al., 2005), a thorough study on the distribution of N and
P pools of the whole ecosystems is still lacking, as vegetation (living or dead biomass)
composes the most active part of the nutrient stocks.
To fill this knowledge gap, here we identified N and P density patterns in China based on
an intensive field investigation, covering all components of the entire ecosystem, including
different plant organs, litter and soil. The present study aims to provide a high-resolution map
of nutrient densities in different ecosystem components and to answer the following questions.
1) How much N and P are stored in different components, i.e., leaf, stem, root, litter and
soil, of terrestrial ecosystems in China?
2) How do different components of N and P pools spatially distribute in China?
**2    Material and methods**
*2.1 Field sampling and nutrient density calculation*
Forest, shrublands and grasslands constitute major vegetation type groups in China.
Focusing primarily on these three groups, a nationwide, methodologically consistent field
investigation was conducted in June and September, 2011-2015.
In total, 4175 sites, including 2385 forest, 1069 shrubland and 721 grassland sites, were
investigated. At each site, one $20 \times 50$ m² plot was set for forests, three replicated $5 \times 5$ m² plots
were set for shrublands, and ten $1 \times 1$ m² plots were established for grasslands. Species
composition and abundance were investigated in plots. Height (for trees, shrubs and herbs),



diameter at breast height (DBH, at height 130 cm) (for trees), basal diameter (for shrubs) and
crown width (for shrubs and herbs) were measured for all plant individuals in the plots (Tang
et al., 2018a).

Leaves, stems (woody stems) and roots (without distinguishing coarse and fine roots) were

sampled for the five top dominant tree and shrub species, and above- and belowground parts
were sampled for dominant herb species. Soil was sampled at the depths of 0–10, 10–20, 20–
30, 30–50, and 50–100 cm with at least five replications per site to measure nutrient
concentrations and bulk density after removing roots and gravels. Litter was sampled in at least
three $1 \times 1$ m$^2$ quadrats per site (for detailed survey protocol, see Tang et al., 2018a).

All samples were transported to laboratory, dried and measured. N concentrations of all

samples were measured by a C/N analyzer (PE-2400 II; Perkin-Elmer, Boston, USA), while P
concentrations were measured using the molybdate/ascorbic acid method after $H_2SO_4$-$H_2O_2$
digestion. For the three organs, the community-level N or P density was the cumulative sum of
the products of the corresponding biomass density (i.e. biomass per area, Mg ha$^{-1}$) and
community-level concentrations for each co-occurring species. For detailed calculation of
species biomass and community-level concentrations in each site, please referred to Tang et al
(2018b).

$$N(P) = \sum B_i \times \theta_i \qquad (1)$$

$N(P)$ represents the community-level N or P density (Mg ha$^{-1}$); $B_i$ is the biomass density

of a specific organ of the $i^{th}$ plant species in one site, where the plant organ biomass was
estimated by allometric equations or harvesting; $\theta_i$ represents the N or P concentration (g kg$^{-}$
$^1$) of the same organ of the $i^{th}$ plant species in that site. Allometric equation methods were
adapted to trees and some shrubs (tree-like shrubs and xeric shrubs) for biomass estimation,



while the biomass of grass-like shrubs and herbs were obtained by direct harvesting. Litter N
or P density was litter biomass density (by harvesting) multiplied by litter N or P concentration
of each sampling site. The soil N or P density was calculated to a depth of one metre. Soil N or
P concentration and bulk density were measured at different depths (0–10, 10–20, 20–30, 30–
50, and 50–100 cm) to determine the community-level soil N or P density using Equation (2):

$SOND(SOPD) = \sum(1 - \delta_i) \times \rho_i \times C_i \times T_i/10$                    (2)

where $SOND(SOPD)$ is the total N or P density of the soil (Mg ha⁻¹) in the $i^{th}$ layer (0-

10, 10-20, 20-30, 30-50 and 50-100 cm), $\delta_i$ is the volume percentage of gravel with a diameter >
2mm, $\rho_i$ is the bulk density (g cm⁻³), $C_i$ is the soil N or P concentration (g kg⁻¹), and $T_i$ is
the depth (cm) of the $i^{th}$ layer. For detailed calculations of species biomass and community-
level concentrations at each site, please refer to previous studies (Tang et al., 2018a, 2018b).

*2.2 Climatic and vegetation data*

The daily meteorological observation data from 2,400 meteorological stations across

China were averaged over the 2011-2015 period to generate a spatial interpolation dataset of
mean annual temperature (MAT) and precipitation (MAP), using a smooth spline function
(McVicar et al., 2007) , with a spatial resolution of 1 km. MAT and MAP of each site were
extracted from this dataset.

Elevation was extracted from GTOPO30 with a spatial resolution of 30 arc-seconds

(http://edc.usgs.gov/products/elevation/gtopo30/gtopo30.html). The mean enhanced vegetation
index (EVI) from June to September during the 2011–2015 period was calculated based on
MOD13A3 data with a resolution of 1 km (https://modis.gsfc.nasa.gov/).

Based on the level II vegetation classification of ChinaCover (Land Cover Atlas of the



People's Republic of China Editorial Board, 2017), we classified the vegetation type groups
into the following 13 Vegetation types: five forest types, i.e., evergreen broadleaf forests,
deciduous broadleaf forests, evergreen needle-leaf forests, deciduous needle-leaf forests,
broadleaf and needle-leaf mixed forests; four shrubland types, i.e., evergreen broadleaf
shrublands, deciduous broadleaf shrublands, evergreen needle-leaf shrublands, and sparse
shrublands; and four grassland types, i.e., meadows, steppes, tussocks, and sparse grasslands.

*2.3 Prediction the nationwide nutrient pools and distribution patterns*
We used back-propagation artificial neural network for nutrient density spatial
interpolating. The input layer contained MAT, MAP, longitude, latitude, elevation, EVI and
vegetation types (as dummy variables). We established one artificial neural network for N and
P in five components, respectively. The observation data were randomly grouped into two
subsets, 90% data for training and the other 10% for validation. When building the artificial
network, we used one and two layers, one to 20 hidden neurons per layer, respectively, to find
out a model configuration with the best predicting ability. The training and testing process were
repeated 100 times for each configuration. The best predicting model was selected according to
the minimal mean root mean square error (RMSE). Then the chosen model was used to predict
the nationwide nutrient distribution in corresponding component for 100 times to obtain the
average conditions.
When modelling the nutrient densities in woody stems, we excluded the four grassland
types. The vegetation N or P density was the sum of all plant organs, and the ecosystem N or P
density was the sum of all components.
All densities were log-transformed based on *e*, and explanatory variables were transformed





using the following equation to ensure they were in the same range before modelling.
$$x_i' = \frac{x_i - min(x)}{max(x) - min(x)} \tag{3}$$
where $x_i$ means the $i^{th}$ value of the environmental variables $x$, and max($x$) and min($x$) represent
the maximum and minimum values of $x$, respectively.
The N and P pools in 13 Vegetation types were estimated, respectively. The N and P pools
were calculated from the predicted nationwide densities. The predicted N and P densities were
in 1 km spatial resolution, so the nutrient stock is the density multiply the grid area (1 km$^2$) for
each grid. The nutrient pools of a given vegetation type equals the sum of stocks of the grids
belonging to that type.

*2.4 Data Model uncertainty and validation*
To evaluate the model performance, we calculated the linear relationship between the observed
validation data (10% of the dataset by random sampling) and predicted data that was estimated
based on training data (90% of the dataset by random sampling) for 100 times with the selected
models for every component. The $R^2$, slopes and intercepts of these relationships were estimated
using standard major axis regression. We also mapped the standard deviations (SDs) of the 100-
time predictions of each component to show the uncertainty of our results in different regions.
All statistical analyses were performed using R 3.6.1 (R Core Team, 2019), artificial
networks were built using *neuralnet* package (Günther and Fritsch, 2010), and standard major
axis regression was conducted using *smatr* package (Warton et al., 2012).

**3    Data accessibility**



The datasets of N and P densities of different ecosystem components, " Patterns of nitrogen and
phosphorus pools in terrestrial ecosystems in China", are available from the Dryad Digital
Repository          (the          pre-publication          sharing          link:
https://datadryad.org/stash/share/78EBjhBqNoam2jOSoO1AXvbZtgIpCTi9eT-eGE7wyOk)
(Zhang et al., 2020).

**4   Results**
*4.1 Site average allocation of nutrient among ecosystem components*
The site averaged N and P densities varied among forests, shrublands and grasslands and
among different tissues (Fig. 1 & 2) according to the measured plot data. In average, leaves and
woody stems in the forests stored more N than those in the shrublands ($11 \pm 10$ (mean $\pm$ SD) $\times$
$10^{-2}$ Mg N ha$^{-1}$ vs. $3.2 \pm 10 \times 10^{-2}$ Mg N ha$^{-1}$ for leaves, and $260 \pm 340 \times 10^{-3}$ Mg N ha$^{-1}$ vs. 5.8
$\pm 11 \times 10^{-3}$ Mg N ha$^{-1}$ for woody stems). Similarly, P densities were higher in the forests leaves
and woody stems than those in the shrublands ($12 \pm 13 \times 10^{-3}$ Mg P ha$^{-1}$ vs. $2.9 \pm 6.1 \times 10^{-3}$ Mg
P ha$^{-1}$ for leaves and $52 \pm 110 \times 10^{-3}$ Mg P ha$^{-1}$ vs. $4.4 \pm 11 \times 10^{-3}$ Mg P ha$^{-1}$ for woody stems).
than those in shrublands ($3.2 \pm 10 \times 10^{-2}$ Mg N ha$^{-1}$ and $2.9 \pm 6.1 \times 10^{-3}$ Mg P ha$^{-1}$ for leaves;
$5.8 \pm 11 \times 10^{-3}$ Mg N ha$^{-1}$ and $4.4 \pm 11 \times 10^{-3}$ Mg P ha$^{-1}$ for woody stems) and grasslands (2.7
$\pm 2.4 \times 10^{-2}$ Mg N ha$^{-1}$ and $2.7 \pm 2.9 \times 10^{-3}$ Mg P ha$^{-1}$ for leaves). However, the root N and P
densities in forests ($1.3 \pm 1.6 \times 10^{-1}$ Mg N ha$^{-1}$ and $1.8 \pm 2.8 \times 10^{-2}$ Mg P ha$^{-1}$) and grasslands
($1.9 \pm 1.7 \times 10^{-1}$ Mg N ha$^{-1}$ and $1.5 \pm 1.6 \times 10^{-2}$ Mg P ha$^{-1}$) were remarkably higher than in
shrublands ($6.5 \pm 11 \times 10^{-2}$ Mg N ha$^{-1}$ and $6.1 \pm 9.9 \times 10^{-3}$ Mg P ha$^{-1}$).
The site-averaged litter N densities in forests, shrublands and grasslands were $6.3 \pm 8.1 \times$
$10^{-2}$ Mg N ha$^{-1}$, $3.2 \pm 4.1 \times 10^{-2}$ Mg N ha$^{-1}$ and $5.5 \pm 9.3 \times 10^{-3}$ Mg N ha$^{-1}$, respectively. The



site-averaged litter P densities in forests, shrublands and grasslands were $5.3 \pm 9.9 \times 10^{-3}$ Mg P
ha$^{-1}$, $2.2 \pm 2.9 \times 10^{-3}$ Mg P ha$^{-1}$ and $4.14 \pm 7.1 \times 10^{-4}$ Mg P ha$^{-1}$, respectively.
The site-averaged soil N densities in forests, shrublands and grasslands were $11.2 \pm 9.2$
Mg N ha$^{-1}$, $9.4 \pm 7.8$ Mg N ha$^{-1}$ and $9.9 \pm 8.9$ Mg N ha$^{-1}$, respectively. The site-averaged soil P
densities were $4.6 \pm 4.2$ Mg P ha$^{-1}$ in forest, $4.0 \pm 3.0$ Mg P ha$^{-1}$ in shrublands and $4.1 \pm 2.7$ Mg
P ha$^{-1}$ in grasslands.
Both belowground vegetation N and P densities were higher than aboveground in
shrublands and grasslands. By contrast, this condition was reversed in forests (Fig. 3). Among
various forest types, deciduous needle-leaf forests held the highest aboveground N and P
densities. Evergreen needle-leaf forests held the lowest vegetation N density and evergreen
broadleaf forests owned lowest P density. For grassland types, the density allocation varied
markedly. Meadows and steppes held higher N and P densities in belowground biomass than
tussocks and sparse grasslands, whereas these four grasslands types had relatively approximate
nutrient densities in aboveground biomass. Shrublands possessed the lowest vegetation N and
P densities among three vegetation groups. Sparse shrublands owned the lowest vegetation
nutrient densities and soil N density but the highest soil P density among four shrubland types.

*4.2 Mapping of N and P densities in China's terrestrial ecosystems*
All models of the N and P densities of different components performed well (Fig. 4),
especially those for the woody stems ($R^2 = 0.81$ and 0.69 for N and P densities, respectively)
and litter ($R^2 = 0.66$ and 0.62 for N and P densities, respectively). SDs of N and P densities were
relatively higher in western and northeastern China, with values > 5 (Fig. 5k–t). For example,
the predictions of litter N (Fig. 5q) and P (Fig. 5r) showed larger SDs in western Xinjiang and



Tibet.

The leaf N density was high in southern and eastern China, but low in northern and western

China. It was especially high in the Changbai Mountains, the southern Tibet and the southeast
coastal areas (Fig. 5), with a density of >0.1 Mg N ha$^{-1}$. In comparison, it was low in the northern
Xinjiang and northern Inner Mongolia (< 0.01 Mg N ha$^{-1}$). The woody stem and litter N
densities showed the similar patterns to those of the leaves, whereas that in roots was high in
the Mount Tianshan, Mount Alta, Qinghai-Tibetan Plateau, northeastern mountainous area and
the Inner Mongolia steppe (Fig. 5). The vegetation N density was relatively high in eastern
China, Qinghai-Tibetan Plateau, Mount Tianshan and Mount Alta, ranging from 0.5 to 2.5 Mg
N ha$^{-1}$. The soil and ecosystem N densities were low in northern China except the Changbai
Mountains, Mount Tianshan and Mount Alta, but high in the eastern Qinghai-Tibetan Plateau
and the Yunnan Province (Fig. 6).

The P densities in leaves, woody stems, roots and litter showed similar patterns to the N

densities in the corresponding components, respectively. However, soil and ecosystem P
densities were high in western and northern China but low in eastern and southern China, but
low at high altitudes in the Qinghai-Tibetan Plateau (Fig. 5 & 6).

*4.3 N and P pools in China's terrestrial ecosystems*

In total, the terrestrial ecosystems in China stored 7665.62 × 10$^6$ Mg N, with 2632.80 ×

10$^6$ Mg N, 830.24 × 10$^6$ Mg N and 4202.58 × 10$^6$ Mg N stored in the forests, shrublands and
grasslands, respectively (Table 1). Vegetation, litter and soil stored 216.17 × 10$^6$ Mg N (2.82%),
14.92 × 10$^6$ Mg N (0.19%) and 7434.53 × 10$^6$ Mg N (96.99%), respectively.

China's terrestrial ecosystems stored 3852.66 × 10$^6$ Mg P, with 1037.34 × 10$^6$ Mg P, 361.62



× 10⁶ Mg P and 2453.70 × 10⁶ Mg P stored in the forest, shrublands and grasslands, respectively.
Vegetation, litter and soil accounted for 28.88 × 10⁶ Mg P (0.75%), 2.14 × 10⁶ Mg P (0.06%)
and 3821.64 × 10⁶ Mg P (99.19%), respectively.

Meanwhile, N and P stocks among plant organs showed different allocation patterns (Table

2). Compared with the other two vegetation type groups, forests allocated the majority of N and
P to the stem pool (59.29× 10⁶ Mg N and 13.81× 10⁶ Mg P), followed by the root pool (28.55×
10⁶ Mg N and 5.53× 10⁶ Mg P) and leaf pool (23.84× 10⁶ Mg N and 2.49× 10⁶ Mg P). However,
the root pools in shrublands and grasslands held the most of N and P (4.44× 10⁶ Mg N and
0.38× 10⁶ Mg P for shrublands, and 91.22× 10⁶ Mg N and 5.55× 10⁶ Mg P for grasslands).

Among 4 grassland types, steppe had the largest N stock (1599.47× 10⁶ Mg N), and sparse

grasslands had the largest P stock (1578.83× 10⁶ Mg P) taking the ecosystem as a whole.
Deciduous broadleaf shrublands owned the largest N and P stocks considering the whole
ecosystem (605.09× 10⁶ Mg N and 211.15× 10⁶ Mg P) as well as in vegetation (5.30× 10⁶ Mg
N and 0.58× 10⁶ Mg P), compared with the other 3 shrubland types. The largest ecosystem N
and P stocks across all 13 vegetation types appeared in evergreen needle-leaf forests (43.43×
10⁶ Mg N and 8.37× 10⁶ Mg).

**5   Discussion**
*5.1 Performance and uncertainty of density models*

The accuracy of the models varied among different components. Soil interpolation models

showed poorest accuracy ($R^2$=0.38 for N and 0.27 for P) among these models, partly because
that soil N and P were more stable than those in the plants and litters (Matamala et al., 2008)
and that soil nutrient exchange and storage were largely controlled by geochemical and

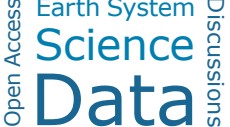

geophysical processes (Doetterl et al., 2015), which are not considered in our models. The
models preformed best for the stem N and P, because woody stems occupied the most biomass
in the forest and shrublands (stem biomass/vegetation biomass were 0.68 and 0.48 for forest
and shrublands, respectively). Climate variables could affect vegetation growth and biomass
accumulation, and the variation in stem biomass could be the most direct reflection (Jozsa and
Powell, 1987; Kirilenko and Sedjo, 2007; Poudel et al., 2011).
The predicted SDs were relatively higher in high-latitudes and high-altitudes, such as the
northeastern mountainous area and the Qinghai-Tibet Plateau, probably because of the lower
sampling density. Meanwhile, the temperature in these regions was about the lower limit of the
temperature range in our dataset, which could consequently lead to the weaker validity of the
prediction results in such cold regions.

*5.2 Potential driving factors of the N and P densities in various components*
The distribution and allocation of N and P pools in ecosystems were largely determined
by vegetation types and climate. The difference in the spatial patterns of nutrient pools could
reflect the spatial variation in local vegetation. For example, it is obvious that the regions
covered by forests tend to have higher the aboveground nutrient densities than those covered
by other types, while the regions covered by sparse shrublands tend to have the lowest nutrient
densities (Fig. 3). Despite its decisive influences on vegetation types, climate also impacts
greatly on the nutrient utilization strategies of vegetation (Kirilenko and Sedjo, 2007; Poudel et
al., 2011). For example, in southeastern China with higher precipitation and temperature, forests
tend to allot more nutrient to organs related to growth, for example, leaves that perform
photosynthesis and stems that related to resource transport and light competition (Zhang et al.,

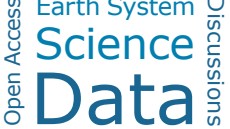

2018). Heat and water are usually limited in the plateau and desert regions in western China,
where shrublands and grasslands are dominant vegetation type groups. More nutrients are
allocated to root systems by dominant plants in such stressful habitats to acquire resources from
soil (Eziz et al., 2017; Kramer-Walter and Laughlin, 2017). Soil nutrient densities were
relatively larger in the plateau and mountainous area in western China, possibly because of the
lower rates of decomposition, mineralization, and nutrient uptake as well as less leaching loss
in high-altitude regions (Bonito et al., 2003; Vincent et al., 2014). However, the distribution
patterns of soil nutrient densities in eastern China were generally consistent with the Soil
Substrate Age hypothesis that the younger and less-leached soil in temperate regions tend to be
more N limited but less P limited than the elder and more-leached soil in tropical and subtropical
regions (Reich and Oleksyn, 2004; Vitousek et al., 2010).

*5.3 Potential applications of the data*
Atmospheric $CO_2$ enrichment trend was undoubtable, but how this procedure will develop is
still unclear (Fatichi et al., 2019). A number of previous studies proved that global carbon cycle
models would produce remarkable bias if overlooking the coupled nutrient cycle (Fleischer et
al., 2019; Hungate et al., 2003; Thornton et al., 2007). However, high-resolution and accurate
ecosystem nutrient datasets were unattainable and hard to be modeled without enormous field
investigation basis. This study relied on nationwide field survey data, providing comprehensive
N and P density datasets of different ecosystem components. Based on the present dataset,
enhancement could be made in various ecosystem research aspects.
First and foremost, the dataset could facilitate the improvement in the prediction of large-
scale terrestrial C budget, thereby to better understand patterns and mechanisms of C cycle as



well as the future trend of climate change (Le Quéré et al., 2018). Numerous projections of
future C sequestration overestimated the amount of C fixed by vegetation due to the neglect of
nutrient limitation (Cooper et al., 2002; Cramer et al., 2001). Global C cycling models coupled
with nutrient cycle could make more accurate predictions of carbon dynamics. Moreover, our
dataset illustrated N and P densities of major ecosystem components and vegetation types at a
high spatial resolution for the first time, which could help identify C and nutrient allocation
patterns from the tissue level to the community level, especially for vegetation organs which
still lack large-scale nutrient datasets.
In addition, large-scale N and P pool spatial patterns could provide the data references for
the vegetation researches using remote sensing (Jetz et al., 2016). Vegetation nutrient densities
was important traits but hard to be extracted and detected remotely. With the development of
hyperspectral remote sensing technology and theory of spectral diversity, foliar nutrient traits
can be successfully predicted (Skidmore et al., 2010; Wang et al., 2019). However, previous
studies still focused on finer-scale patterns and were constrained by the lack of large-scale field
datasets for uncertainties assessment (Singh et al., 2015). Our nationwide nutrient dataset offers
an opportunity to enlarge the generality of remote-sensing models and algorithms at large scales.

**Funding**
This work was funded by the National Key Research and Development Project
(2019YFA0606602), the National Natural Science Foundation of China (32025025,
31770489, 31988102) and the Strategic Priority Research Programme of the Chinese
Academy of Sciences (XDA27010102, XDA05050000).
**Author Contributions**





Z.T. designed the research. Y.W.Z, Y.G., Y.F., and X.Z. analysed the data. W.X., Y.B., G.Z.,
Z.X. and Z.T. organized the field investigation. Y.W.Z, Y.G., Z.T. wrote the manuscript and
all authors contributed substantially to revisions.

**Competing interests**
The authors declare no competing interests.





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






**Table.1.** N and P stocks of vegetation, litter, soil and total ecosystem in forest, shrublands and grasslands in China.

| Vegetation type group | Vegetation type | Area (10⁶ ha) | N pool (10⁶ Mg) | | | | P pool (10⁶ Mg) | | | |
|---|---|---|---|---|---|---|---|---|---|---|
| | | | Vegetation | Soil | Litter | Ecosystem | Vegetation | Soil | Litter | Ecosystem |
| Forest | EBF | 45.59 | 13.08 | 587.48 | 1.93 | 608.73 | 1.89 | 193.27 | 0.10 | 195.26 |
| | DBF | 91.14 | 43.43 | 665.60 | 4.25 | 713.29 | 8.37 | 277.61 | 0.74 | 286.71 |
| | ENF | 99.97 | 34.93 | 1074.18 | 3.58 | 1112.69 | 5.59 | 377.27 | 0.23 | 383.08 |
| | DNF | 19.79 | 7.53 | 84.76 | 1.73 | 94.03 | 4.61 | 123.33 | 0.75 | 128.70 |
| | MF | 13.25 | 6.47 | 96.95 | 0.65 | 104.07 | 1.38 | 42.08 | 0.13 | 43.59 |
| | *subtotal* | *269.75* | *111.69* | *2508.98* | *12.14* | *2632.80* | *21.84* | *1013.55* | *1.96* | *1037.34* |
| Shrubland | EBS | 21.65 | 1.56 | 160.54 | 0.61 | 162.70 | 0.20 | 52.95 | 0.03 | 53.18 |
| | DBS | 63.94 | 5.30 | 598.39 | 1.40 | 605.09 | 0.58 | 220.48 | 0.09 | 221.15 |
| | ENS | 1.36 | 0.06 | 13.29 | 0.01 | 13.36 | 0.008 | 5.40 | 0.0006 | 5.41 |
| | SS | 17.35 | 0.22 | 48.66 | 0.21 | 49.10 | 0.02 | 81.85 | 0.01 | 81.88 |
| | *subtotal* | *104.31* | *7.14* | *820.88* | *2.23* | *830.24* | *0.80* | *360.69* | *0.13* | *361.62* |
| Grassland | ME | 59.62 | 17.87 | 994.70 | 0.13 | 1012.70 | 1.33 | 217.20 | 0.005 | 218.54 |
| | ST | 190.08 | 36.31 | 1562.94 | 0.22 | 1599.47 | 2.32 | 569.27 | 0.02 | 571.61 |
| | TU | 24.39 | 2.39 | 171.02 | 0.10 | 173.51 | 0.26 | 84.44 | 0.01 | 84.71 |
| | SG | 139.27 | 40.78 | 1376.02 | 0.09 | 1416.89 | 2.33 | 1576.48 | 0.02 | 1578.83 |
| | *subtotal* | *413.35* | *97.35* | *4104.68* | *0.55* | *4202.58* | *6.24* | *2447.41* | *0.05* | *2453.70* |
| Total | | *787.4* | *216.17* | *7434.53* | *14.92* | *7665.62* | *28.88* | *3821.64* | *2.14* | *3852.66* |

EBF, evergreen broadleaf forest; DBF, deciduous broadleaf forest; ENF, evergreen needle-leaf forest; DNF, deciduous needle-
leaf forest; MF, broadleaf and needle-leaf forest; EBS, evergreen broadleaf shrub; DBS, deciduous broadleaf shrub; ENS,
evergreen needle-leaf shrub; SS, sparse shrub; ME, meadow; ST, steppe; TU, tussock; and SG, sparse grassland.



**Table.2.** N and P stocks of plant organs (leaf, stem and root) in forest, shrublands and grasslands in China.

| Vegetation type group | Vegetation type | Area (10⁶ ha) | N pool (10⁶ Mg) | | | P pool (10⁶ Mg) | | |
|---|---|---|---|---|---|---|---|---|
| | | | Leaf | Stem | Root | Leaf | Stem | Root |
| Forest | EBF | 45.59 | 3.849 | 10.863 | 4.614 | 0.273 | 1.308 | 0.313 |
| | DBF | 91.14 | 6.820 | 23.289 | 13.322 | 0.555 | 4.680 | 3.133 |
| | ENF | 99.97 | 10.185 | 17.090 | 7.653 | 1.256 | 3.205 | 1.125 |
| | DNF | 19.79 | 1.661 | 4.336 | 1.535 | 0.292 | 3.691 | 0.631 |
| | MF | 13.25 | 1.326 | 3.714 | 1.428 | 0.117 | 0.931 | 0.328 |
| | *subtotal* | *269.75* | *23.841* | *59.293* | *28.552* | *2.493* | *13.814* | *5.531* |
| Shrubland | EBS | 21.65 | 0.632 | 0.051 | 0.872 | 0.045 | 0.074 | 0.083 |
| | DBS | 63.94 | 1.682 | 0.205 | 3.413 | 0.124 | 0.164 | 0.290 |
| | ENS | 1.36 | 0.037 | 0.001 | 0.022 | 0.005 | 0.0001 | 0.003 |
| | SS | 17.35 | 0.070 | 0.021 | 0.129 | 0.005 | 0.005 | 0.006 |
| | *subtotal* | *104.31* | *2.420* | *0.279* | *4.436* | *0.179* | *0.243* | *0.382* |
| Grassland | ME | 59.62 | 1.181 | 0 | 16.687 | 0.121 | 0 | 1.213 |
| | ST | 190.08 | 2.818 | 0 | 33.492 | 0.261 | 0 | 2.055 |
| | TU | 24.39 | 0.559 | 0 | 1.830 | 0.058 | 0 | 0.201 |
| | SG | 139.27 | 1.573 | 0 | 39.211 | 0.244 | 0 | 2.084 |
| | *subtotal* | *413.35* | *6.132* | *0* | *91.220* | *0.685* | *0* | *5.553* |
| Total | | 787.4 | 32.394 | 59.571 | 124.209 | 3.357 | 14.057 | 11.466 |

See table 1 for abbreviations.

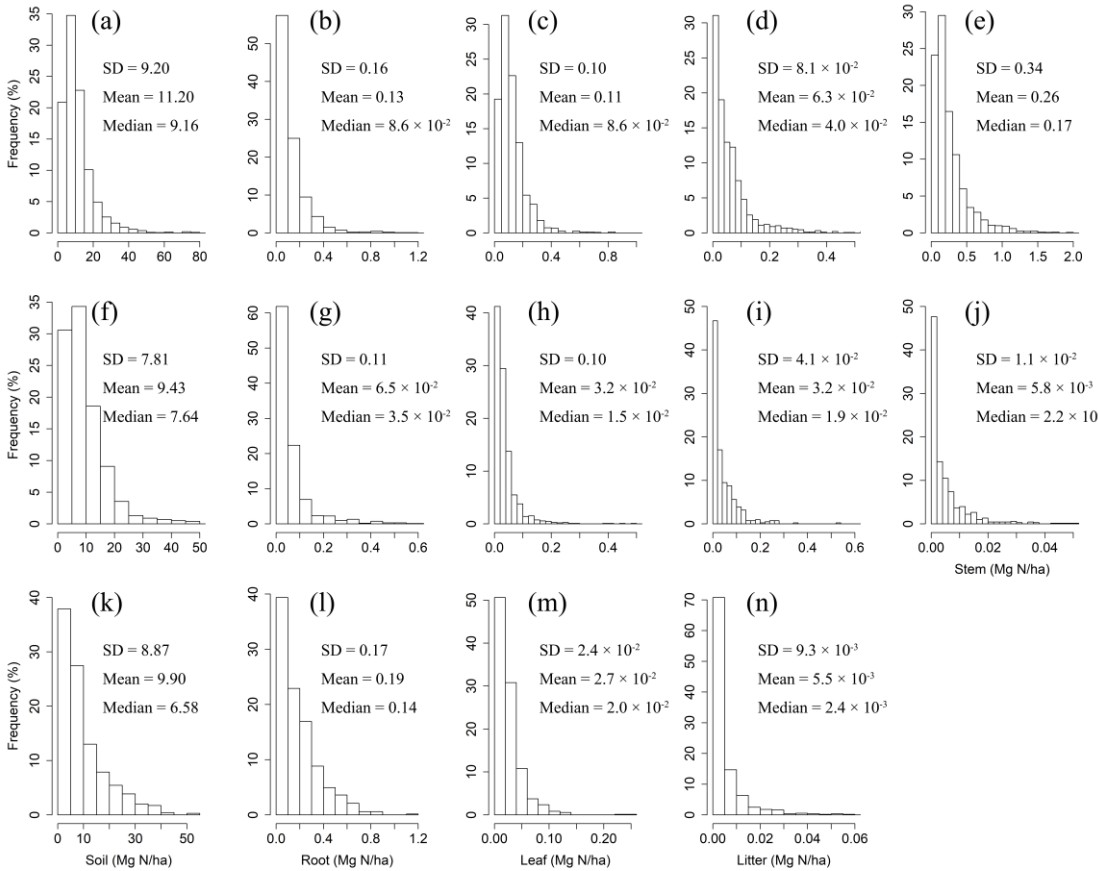

**Fig. 1.** Frequency distributions of N densities in soil, roots, leaves, litter and woody stems in forests (a–e), shrublands (f–j) and grasslands (k–n) in China.



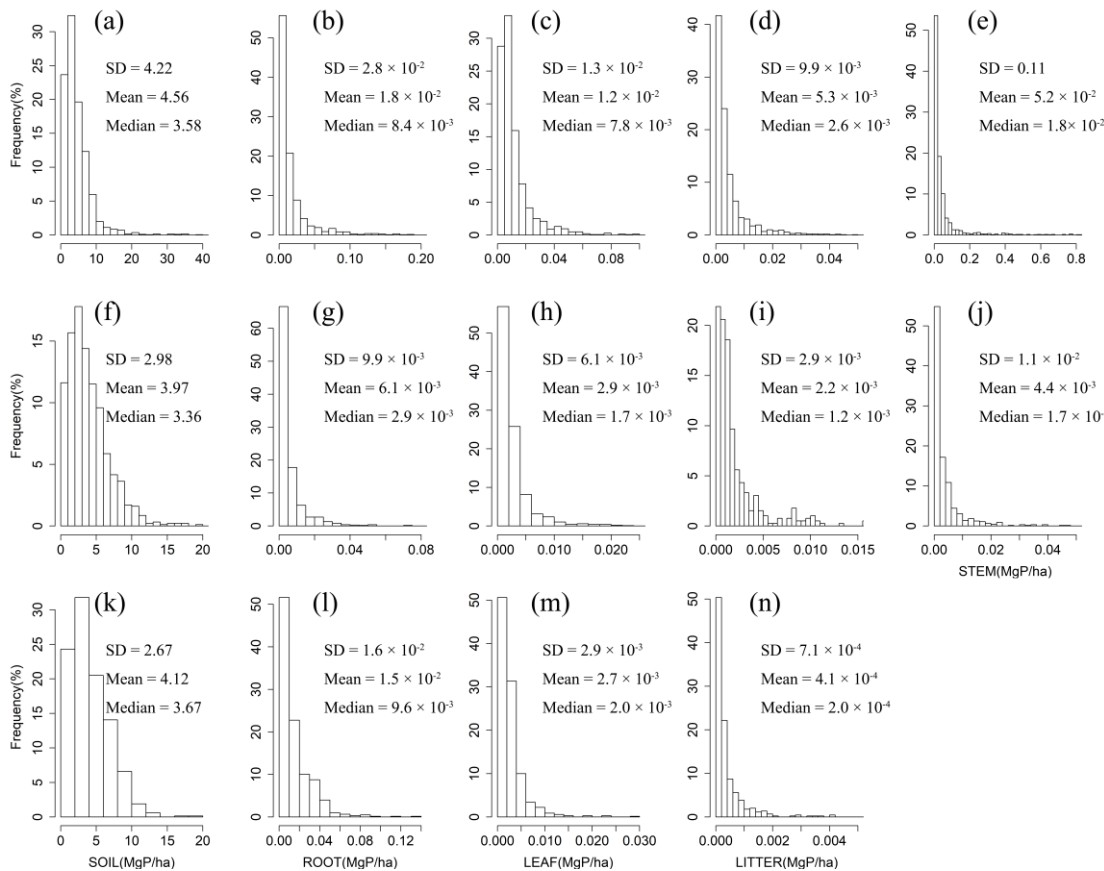

**Fig. 2.** Frequency distributions of P densities in soil, roots, leaves, litter and woody stems in forests (a–e), shrublands (f–j) and grasslands (k–n) in China.

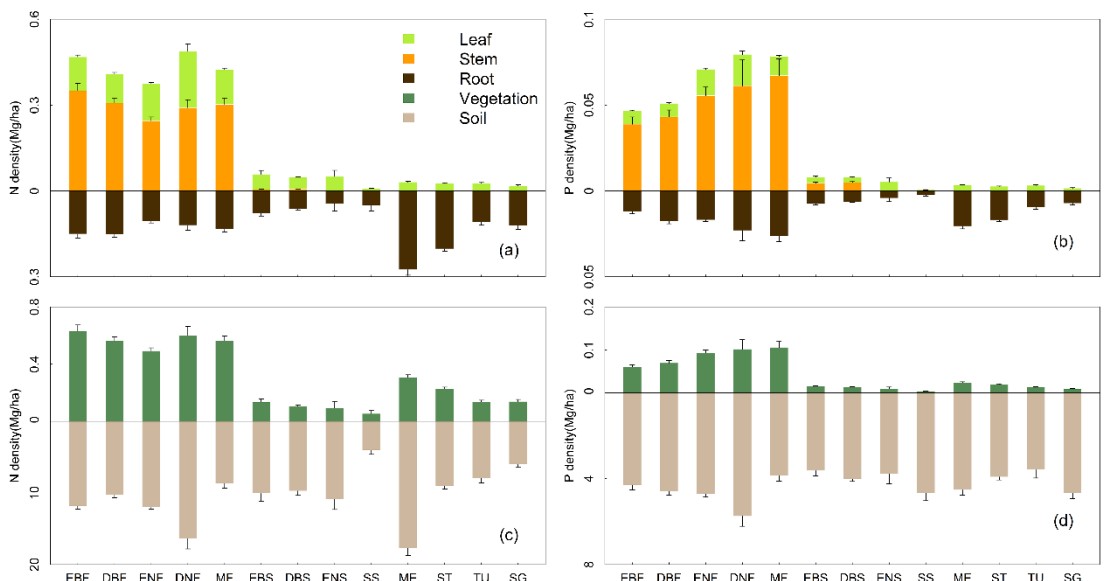

**Fig. 3.** N and P density allocations among leaf, stem and root (a & b) and between vegetation
and soil (c & d) in 13 Vegetation types. See table 1 for abbreviations. The error bar represents
standard error. Notice that the y axes above and below zero are disproportionate.



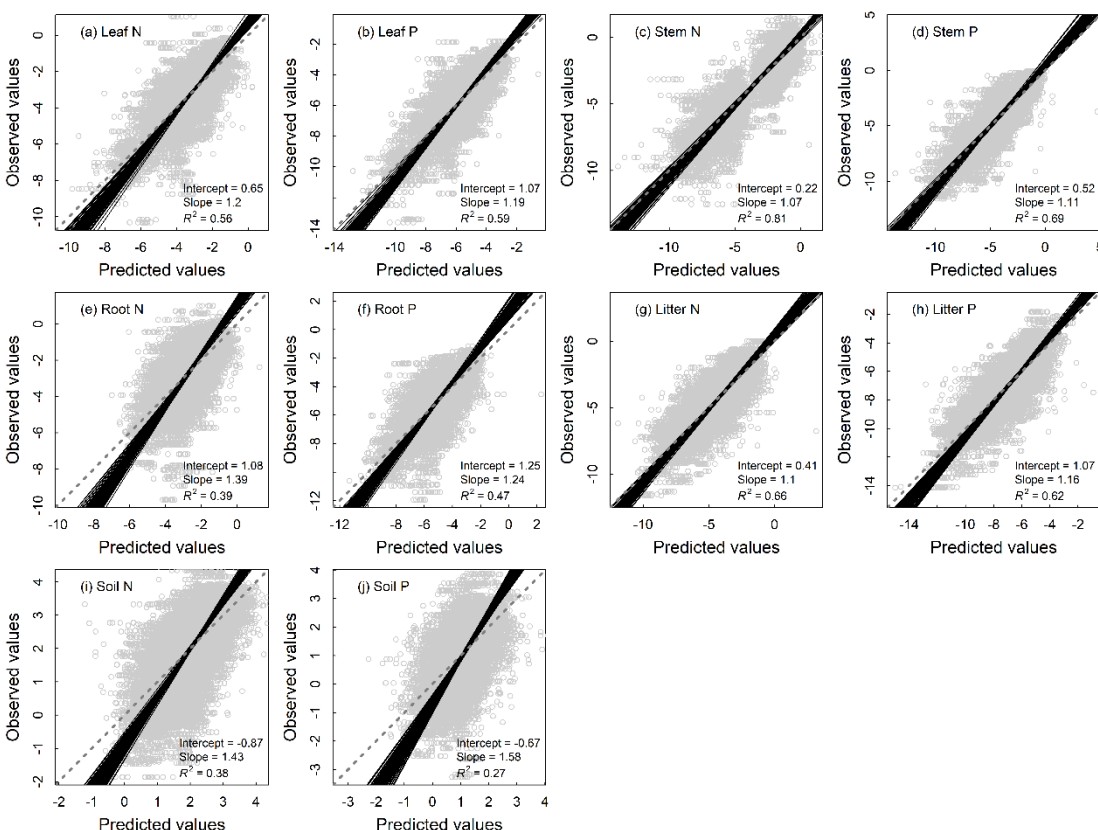

**Fig. 4.** Fitting performance of artificial neural network models for different components of terrestrial ecosystems in China based on 100 times of replications with the 10% validation data. Solid lines represent all the fitting lines by standard major axis regression, and the displayed parameters stand for the average conditions. The dashed line denotes the 1:1 line.



594

**Fig. 5.** Predicted spatial patterns of N and P densities with a resolution of 1 km (a–j) and their
prediction standard deviations (SDs) (k–t) in each component of terrestrial ecosystems in China
based on 100 replications. The topographic map of China (u) is also shown.

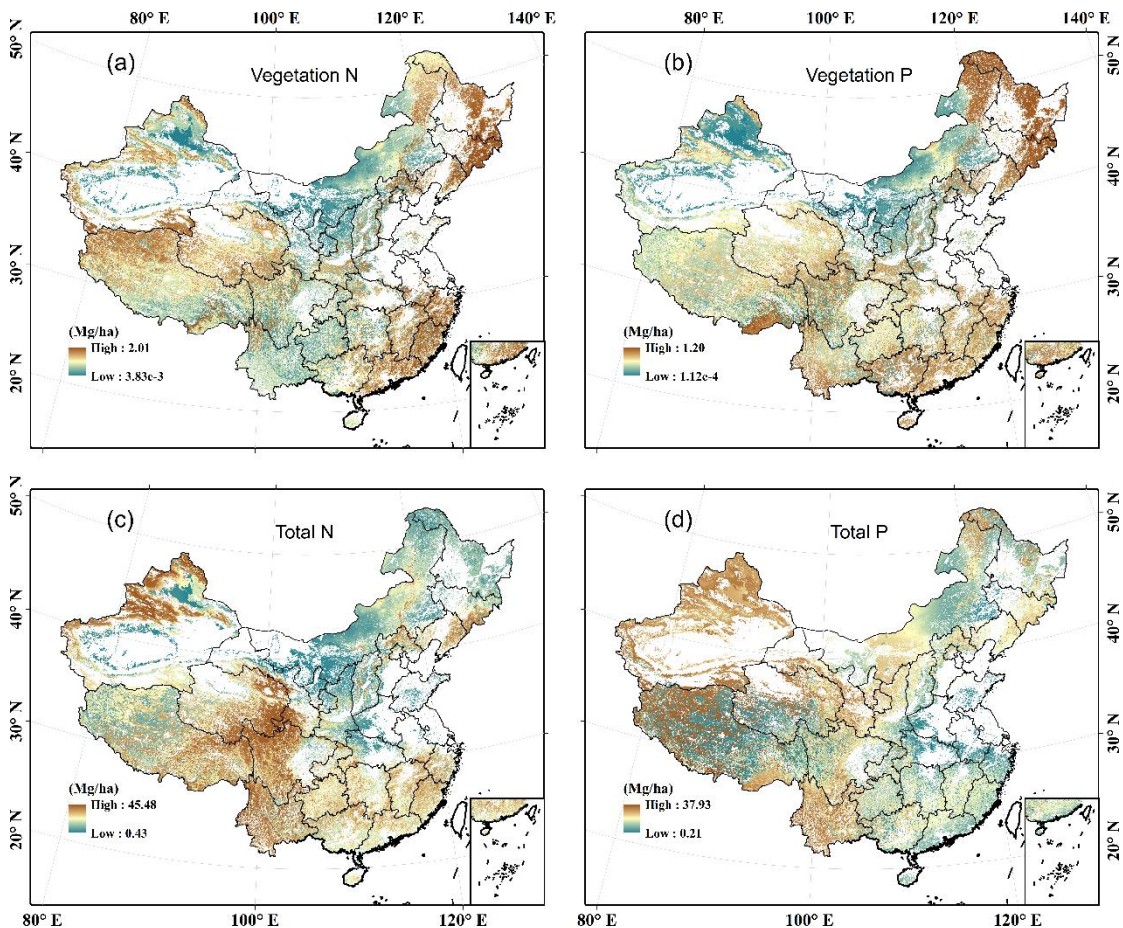

598

**Fig. 6.** Spatial patterns of N and P densities with a resolution of 1 km in vegetation (a & b, the

sum of leaves, stems and roots) and ecosystems (c and d, the sum of leaves, stems, roots, litter

and soil) of terrestrial ecosystems in China.

602