# Peer review of "Patterns of nitrogen and phosphorus pools in terrestrial ecosystems in China"

_Earth System Science Data, 2020_

## Author Comment (AC1)

**Earth System Science Data**

**RE: Manuscript ID: essd-2020-398**

**Point-for-point responses to the comments from Reviewer 1**

Note: texts in black are the comments, and texts in blue are our responses.

We appreciate your constructive comments on our manuscript. We carefully considered each of them and revised the manuscript accordingly. We hope that you will find the revisions satisfactory.

**General comment**

Yi-Wei Zhang et al. presented a data analysis study for terrestrial ecosystem N and P pools over China. The data collection, model fitting, regional and pft level aggregation and analysis are well done. The presentation is smooth.

Response: Thank you very much for reviewing our manuscript. We appreciate your kind praise for our study!

Below are my major suggestions and specific comments.

1 Root N, P and Soil N, P model fitting

Root and soil N, P models underperformed (e.g., R2 0.27~0.47), in comparison with models of other plant components (e.g., R2=0.56-0.81). I would suggest 1) trying more complex neural network models (more layers or more nodes within each layer) 2) trying different types of ML models (e.g., random forest, support vector regression) 3) including more explaining variables besides MAT, MAP, elevation, and PFT. For example, N/P deposition, land use history, soil order, GPP and so on.

Response: Thank you for this comment. After comparing several models, we have adopted the random forest method to reach a better model performance. Furthermore, we fitted models for each soil layer, respectively, instead of the sum of all layers in the previous manuscript. In the revised results,  $R^2$  of the models were higher than the previous version, with *ca* 0.6 for root N and P, and *ca* 0.5 for different layers of soil N and P concentration. We did not include other factors because of the lack of N/P deposition and soil order at such fine scale. As we focus on the natural ecosystems, land use history is not considered in this study. The process of random forest modelling was stated as follows (L178-L185):

"We used random forest to predict the nutrient densities and concentrations across China. The predictors included MAT, MAP, longitude, latitude, elevation, EVI and vegetation types (as dummy variables). We established one random forest model for N or P density in each component (in three plant organs, litter and five soil layers), respectively. In each model, six variables were randomly sampled at each split, and 500 trees were grown. Larger values of these parameters did not increase validation R2 obviously. Model prediction were repeated for 100 times to obtain the average results..." For detailed results please see Fig 4-7 in the revised manuscript.

**2 representativeness of data for regional extrapolation**

It will be helpful to show 1) a map that includes the location of all data samples 2) MAT, MAP, elevation ranges for data samples, compared with those variables but

across China. The purposes are to reveal whether the data samples are spatially representative and whether the data reasonably cover the full range of T, P, Elevation so that the spatial extrapolation is reliable (for each vegetation cover).

Response: Thank you for mention this. We revised the manuscript according to this suggestion. Please see Fig S1 the supplement of the revised manuscript for the distribution map of sampling sites. The environmental variables of our sampling sites could generally cover the 99% ranges of the corresponding variables across China. We supplemented the relevant descriptions in the revised manuscript as follows (L164-L168):

"The ranges of these variables of our field sites (EVI:  $0.03 \sim 0.7$ ; elevation: -137 m~5797 m; MAP: 19.8 mm~2316.3 mm; MAT: -5.2 °C~ 26.0 °C) could generally cover the ranges of corresponding variables in the focused vegetation types across China (99% ranges of EVI:  $0.03 \sim 0.6$ ; of elevation: 24 m~5628 m; of MAP: 50.6 mm~2956.5 mm; of MAT: -6.6 °C~ 22.8 °C)."

**3. N, P mass concentration**

This analysis focused on area-based N, P concentrations (g N/m2 of land surface), which do not directly link to ecosystem N/P limitations. And given that the vegetation is not evenly distributed, it will be helpful to also present the mass-based N, P concentrations (e.g., g N/g tissue biomass or soil) that could directly reveal the strength of plant and soil N, P limitation.

Response: Thank you for this suggestion. We revised the manuscript according to your suggestion and supplemented the results about mass-based concentrations in the revised manuscript.

In Methods, we mentioned that "The same procedures were repeated for the prediction of N and P concentrations in different components across China." (L192–193).

In Results, we supplemented that "The N and P concentrations in plant organs and litter were generally higher in northern and western mountain regions, but larger values of the former often occurs in northwestern part of China, while those of the latter often occurs in northeastern part of China (Fig. S6a–h). The spatial patterns of soil nutrient concentrations at different depths were consistent with those of soil nutrient densities (Fig. S6i–r)." (L274–L278). Please see Fig S6 for the detailed results.

**4. N:P stoichiometry**

From an ecosystem N/P limitation perspective, the ratio of N and P within different plant tissues will be more informative than the individual concentrations. I would suggest also showing N:P stoichiometry, e.g., across pfts, leaf vs fine root.

Response: Thank you for this suggestion. We revised the manuscript according to your suggestion and supplemented the prediction maps of N:P in different plant tissues and soil in the revised manuscript.

In Methods, we mentioned that "The spatial pattern of N:P ratio was calculated from the predicted N and P density datasets of the corresponding component." (L194–195).

In Results, we supplemented that "N:P ratio of plant organs and litter showed similar distribution patterns, higher values occurring in southeastern and northwestern China

and Qinghai-Tibetan Plateau (Fig. S7a–d). Soil N:P ratio was higher in northeastern and southern China but lower in northwestern China (Fig. S7e)." (L279–L282). Please see Fig S7 for the detailed results.

**Specific comments:**

L54 independently or jointly L63 allocated to plant

Response: Thank you for the suggestions. We have corrected these in the revised manuscript.

L167 since the model uses re-scaled predictors (eq. 3), it is important the make sure the training data could represent the full climate envelopes over China.

Response: Thanks for this suggestion. We supplemented the descriptions about the ranges of environmental variables of the sampled data. Please see our answer to the major comment 2 above.

**L226 what is site-averaged?**

Response: Thank you for this comment. We corrected this description at L233 and other parts in the revised manuscript:

"The mean litter N densities for forest, shrubland and grassland sites were  $6.1 \pm 7.6 \times 10-2$  Mg N ha-1,  $3.8 \pm 4.6 \times 10-2$  Mg N ha-1 and  $5.5 \pm 9.3 \times 10-3$  Mg N ha-1, respectively..."

**L238 density varied**

**Response: Thanks for this comment. We have made this correction.**

L294 "soil N and P are stable" is not a convincing reason why soil models underperformed. In contrast, one would expect that stable N P pools shall be better modeled by long-term climatology, compared with e.g, seasonally changed leaf N/P concentrations.

Response: Thank you for the suggestion. Besides other reasons, we think the influence of soil properties (such geological conditions, soil age and parent material), which was not included our analysis, may weaken the soil models. This can be evidenced by the decreasing  $R^2$  of the models with soil depth. We have changed the text in this part of the manuscript (L310-L315).

"Models for soil showed relatively poorer accuracy than models for plant organs and litter (Fig. 4 & 5), partly because that soil N and P were largely influenced by geological conditions, soil age and parent material (Buol and Eswaran, 1999; Doetterl et al., 2015; Gray and Murphy, 2002), which were not included in our analysis because of the limited data availability. The can be evidenced by the decreasing validation R2 of the models for soil N densities and P densities and concentrations with soil depths (Fig. 5 and S3)." L309 this section needs more quantitative evidence for drivers that are included in this study (e.g., T, P, elevation) and should consider including potential drivers that are discussed if spatial data are available (e.g., soil age, soil order).

Response: Thank you for this comment. We supplemented the contributions of each variables in appendix fig S8-S11. Although we did not include soil age and soil order, we discussed the potential contribution of these variables in our models and possible drivers in this section

L352-L354: "These influences were reflected in our models (Fig. S8-S11). In the models for plant organs and litter, vegetation types and climate variables showed higher relative importance."

L358-L365: "Spatial variables, longitude and latitude, also held high importance, especially in the models for soil nutrients. On the one hand, it may result from their tight links with climate conditions. On the other hand, it may imply the influence of spatial correlation on nutrient pools. The effects of elevation and spatial variables were obvious from the prediction maps. There were relatively larger values of soil nutrient densities in the plateau and mountainous area in western China, possibly because of the lower rates of decomposition, mineralization, and nutrient input as well as less leaching loss in high-altitude regions (Bonito et al., 2003; Vincent et al., 2014)."

L369-L372: "Additionally, such patterns reflect that the factors not investigated in this study, such as soil age and parent material, could contribute to the patterns of nutrient pools, which should be considered in future researches as potential drivers (Porder and Chadwick, 2009; Augusto et al., 2017)."

**Tables and figures in the revised manuscript**

| 2 | Table.1. N and P | stocks of vegetation, | litter, soil and total | l ecosystem in forests, | shrublands and | grasslands in China. |
|---|------------------|-----------------------|------------------------|-------------------------|----------------|----------------------|
|---|------------------|-----------------------|------------------------|-------------------------|----------------|----------------------|

| Vegetation | Vegetation | Area          | N pool (Ta)  |        |            |           | D pool (Ta) |        |        |           |
|------------|------------|---------------|--------------|--------|------------|-----------|-------------|--------|--------|-----------|
| type group | type       | $(10^{6} ha)$ | N pool (1g)  |        |            |           | P pool (1g) |        |        |           |
|            |            |               | Vegetation   | Soil   | Litter     | Ecosystem | Vegetation  | Soil   | Litter | Ecosystem |
| Forest     | EBF        | 40.6          | 18.0         | 476.4  | 1.7        | 496.1     | 1.7         | 154.8  | 0.1    | 156.6     |
|            | DBF        | 66.3          | 43.1         | 811.3  | 3.7        | 858.1     | 6.9         | 346.5  | 0.4    | 353.8     |
|            | ENF        | 83.8          | 28.4         | 952.8  | 2.8        | 984.0     | 3.7         | 349.2  | 0.2    | 353.1     |
|            | DNF        | 11.5          | 5.6          | 177.7  | 0.5        | 183.8     | 1.5         | 73.6   | 0.1    | 75.2      |
|            | MF         | 9.6           | 4.6          | 107.6  | 0.5        | 112.8     | 0.9         | 41.5   | 0.1    | 42.4      |
|            | subtotal   | 211.9         | 99 .8 | 2525.8 | 9.3 | 2634.9    | 14.6        | 965.6  | 0.9    | 981.1     |
|            |            |               |              |        |            |           |             |        |        |           |
| Shrubland  | EBS        | 18.7          | 2.1          | 213.6  | 0.5        | 216.2     | 0.2         | 80.9   | < 0.1  | 81.1      |
|            | DBS        | 48.7          | 5.5          | 570.9  | 1.2        | 577.6     | 0.5         | 233.6  | 0.1    | 234.2     |
|            | ENS        | 1.0           | 0.1          | 12.4   | < 0.1      | 12.5      | < 0.1       | 4.9    | < 0.1  | 4.9       |
|            | SS         | 11.9          | 0.5          | 66.1   | 0.1        | 66.7      | < 0.1       | 61.6   | < 0.1  | 61.6      |
|            | subtotal   | 80.3          | 8.1          | 863.0  | 1.8        | 873.0     | 0.7         | 381.0  | 0.1    | 381.8     |
|            |            |               |              |        |            |           |             |        |        |           |
| Grassland  | ME         | 44.2          | 11.6         | 806.9  | 0.1        | 818.5     | 0.9         | 247.2  | < 0.1  | 248.0     |
|            | ST         | 137.4         | 21.3         | 1348.5 | 0.3        | 1370.1    | 1.5         | 573.1  | < 0.1  | 574.6     |
|            | TU         | 22.8          | 2.3          | 230.4  | 0.1        | 232.8     | 0.2         | 112.9  | < 0.1  | 113.2     |
|            | SG         | 103.8         | 13.6         | 860.6  | 0.1        | 874.4     | 0.9         | 506.3  | < 0.1  | 507.2     |
|            | subtotal   | 308.2         | 48.8         | 3246.4 | 0.6        | 3295.8    | 3.5         | 1439.5 | < 0.1  | 1443.0    |
| Total      |            | 600.4         | 156.7        | 6635.2 | 11.7       | 6793.1    | 18.8        | 2786.1 | 1.0    | 2806.0    |

- 3 EBF, evergreen broadleaf forest; DBF, deciduous broadleaf forest; ENF, evergreen needle-leaf forest; DNF, deciduous needle-
- 4 leaf forest; MF, broadleaf and needle-leaf forest; EBS, evergreen broadleaf shrub; DBS, deciduous broadleaf shrub; ENS,
- 5 evergreen needle-leaf shrub; SS, sparse shrub; ME, meadow; ST, steppe; TU, tussock; and SG, sparse grassland.

| Vegetation type group | Vegetation type | Area (10 6 ha) | N pool (Tg) |       | P pool (Tg) |       |       |       |
|-----------------------|-----------------|---------------------------|-------------|-------|-------------|-------|-------|-------|
|                       |                 |                           | Leaf        | Stem  | Root        | Leaf  | Stem  | Root  |
| Forest                | EBF             | 40.6                      | 3.9         | 10.1  | 4.0         | 0.3   | 1.0   | 0.3   |
|                       | DBF             | 66.3                      | 6.1         | 26.6  | 10.5        | 0.6   | 4.6   | 1.6   |
|                       | ENF             | 83.8                      | 8.6         | 13.4  | 6.4         | 0.9   | 2.0   | 0.8   |
|                       | DNF             | 11.5                      | 1.3         | 2.9   | 1.4         | 0.2   | 0.9   | 0.3   |
|                       | MF              | 9.6                       | 1.0         | 2.6   | 1.0         | 0.1   | 0.7   | 0.2   |
|                       | subtotal        | 211.9                     | 21.0        | 55.5  | 23.4        | 2.1   | 9.2   | 3.3   |
| Shrubland             | EBS             | 18.7                      | 0.6         | 0.7   | 0.7         |

9 Fig. 1. Frequency distributions of N densities in soil, roots, leaves, litter and woody stems in

10 forests (a-e), shrublands (f-j) and grasslands (k-n) in China.

---

## Author Comment (AC2)

**Earth System Science Data**

**RE: Manuscript ID: essd-2020-398**

**Point-for-point responses to the comments from Reviewer 2**

*Note*: texts in black are the comments, and texts in blue are our responses.

We appreciate your constructive comments on our manuscript. We carefully considered each of them and revised the manuscript accordingly. We hope that you will find the revisions satisfactory.

Zhang et al. mapped distributions of N and P pools in China terrestrial ecosystems, based on the most intensive field measurements in China ever, including all major (semi-)natural ecosystem types and ecosystem components. The study is generally well performed, and the manuscript is well written. I think the paper deserve a publication on Earth System Science Data and would be highly influential one after published. Before its publication, the authors may improve the manuscript by considering my comments and suggestions as follows.

Response: Thank you very much for reviewing our manuscript. We appreciate your helpful comments to improve this manuscript and revised it accordingly.

Major comments

• I think the authors should justify their use of artificial neural network for mapping. This method is a complex one but necessarily be the best one. Did the authors test or use other methods such as random forest?

Response: Thank you for your suggestion. According to your suggestion, we compared three different methods, the artificial neural network, support vector regression and random forest, and RF outperformed the other two. We therefore adopted random forest in this revised version for a better model performance (L178-L185).

"We used random forest to predict the nutrient densities and concentrations across China. The predictors included MAT, MAP, longitude, latitude, elevation, EVI and vegetation types (as dummy variables). We established one random forest model for N or P density in each component (in three plant organs, litter and five soil layers), respectively. In each model, six variables were randomly sampled at each split, and 500 trees were grown. Larger values of these parameters did not increase validation  $R^2$  obviously. Model prediction were repeated for 100 times to obtain the average results..."

Furthermore, we fitted models for each soil layer, respectively, instead of the sum of all layers in the previous manuscript, and  $R^2$  of these models were all around 0.5, much higher than the previous results. For details please see Fig 4-5 in the revised manuscript.

• Ideally, the authors may also show and discuss the relative importance of the predictors in predicting the nutrient densities. This will help readers to build a more mechanistic view of the patterns. Not sure whether neutral network can do this.

Response: Thank you for this comment. We analyzed the relative importance of variables in methods (L190-L192).

"We estimated the relative importance of predictors using the increase in node purity for the splitting variable, which was measured by the reduction in residual sum of squares."

The relative importance was discussion at L352–L365:

"These influences were reflected in our models (Fig. S8-S11). In the models for plant organs and litter, vegetation types and climate variables showed higher relative importance. Heat and water are usually limited in the plateau and desert regions in western China, where shrublands and grasslands are dominant vegetation type groups. More nutrients are allocated to root systems by dominant plants in such stressful habitats to acquire resources from soil (Eziz et al., 2017; Kramer-Walter and Laughlin, 2017). Spatial variables, longitude and latitude, also held high importance, especially in the models for soil nutrients. On the one hand, it may result from their tight links with climate conditions. On the other hand, it may imply the influence of spatial correlation on nutrient pools. The effects of elevation and spatial variables were obvious from the prediction maps. There were relatively larger values of soil nutrient densities in the plateau and mountainous area in western China, possibly because of the lower rates of decomposition, mineralization, and nutrient input as well as less leaching loss in high-altitude regions (Bonito et al., 2003; Vincent et al., 2014)." For detailed results please see Fig S8-S11 in the supplement of the revised manuscript.

• While I agree with the authors' argument that "the first time, we mapped N and P densities of leaves, woody stems, roots, litter and soil in forest, shrubland and grassland ecosystems across China", there are some previous estimates of nutrient stocks in China, maybe only for one ecosystem component or one nutrient. I think a comparison of the authors' estimates with previous estimates, e.g. Tian et al. (2010), would benefits the study. It will make the study well in context of previous studies, and will also show how the estimates are improved compared to previous estimates.

Tian, H., Chen, G., Zhang, C., Melillo, J.M. & Hall, C.A. (2010). Pattern and variation of C: N: P ratios in China's soils: a synthesis of observational data. *Biogeochemistry*, 98, 139-151.

Response: Thank you for this suggestion. We compared the previous estimation of N and P pools with our results in the section *5.2 Nutrient pools in terrestrial ecosystems in China* (L327-L341):

"Previous researches have estimated N and P stocks in soil across China. For example, Shangguan et al (2013) estimated that the storage of soil total N and P in the upper 1m of soil in China were 6.6 and 4.5 Pg. Yang et al (2007) estimated China's average density of soil N at a depth of one meter which was 0.84kg m-2 and the soil N stock was 7.4 Pg. Zhang et al (2005) investigated soil total P pool at a depth of 50 cm in China and concluded that the soil stock was 3.5 Pg with the total P density of soil  $8.3 \times 102$  g/m3. Our estimation of the soil N pool in China (6.6Pg) agreed with Shangguan et al (2013), but the estimated soil P pool (2.8Pg) was lower than the results of aforementioned studies. The mean soil N:P ratio in our study (2.5 of the predicted dataset and 2.1 of the training dataset) was lower than the result of Tian et al (2010), 5.2, while the spatial patterns in both studies are similar. Other than those researches focusing on soil, Xu et al (2020) estimated China's N storage by calculating the mean N densities of vegetation and soil from different ecoregions, and the reported that there were 10.43 Pg N in China's ecosystem, 10.14 Pg N in top 1 m soil and 0.29 Pg N in vegetation, both higher than our results (6.6 Pg N in soil and 0.16 Pg N in vegetation)."

**Minor comments**

L18-19: "the limitation of these two" may be changed to "their limitations". L26-31: the numbers are unreadable. Mg is million gram? Given the use of 106, you may use bigger units (e.g., Tg).

L49: here you may also cite Sun, Y., Peng, S., Goll, D.S., Ciais, P., Guenet, B., Guimberteau, M. *et al.* (2017). Diagnosing phosphorus limitations in natural terrestrial ecosystems in carbon cycle models. *Earth's Future*, 5, 730-749. L91-92: Not very clear. Du et al. (2020) showed either N or P limitation. If you mean ubiquitous limitation by N and P, you may refer to Elser et al. (2007), LeBauer and Treseder, K.K. (2008), Augusto et al. (2017), and more recently Hou et al. (2020). Similarly, L46-60 may cite more recent papers on the topic to reflect recent progresses in the field.

LeBauer, D.S. & Treseder, K.K. (2008). Nitrogen limitation of net primary productivity in terrestrial ecosystems is globally distributed. *Ecology*, 89, 371-379. Hou, E., Luo, Y., Kuang, Y., Chen, C., Lu, X., Jiang, L. *et al.* (2020). Global meta-analysis shows pervasive phosphorus limitation of aboveground plant production in natural terrestrial ecosystems. *Nature Communications*, 11, 637.

Augusto, L., Achat, D.L., Jonard, M., Vidal, D. & Ringeval, B. (2017). Soil parent material $\hat{a} \in \bullet$  a major driver of plant nutrient limitations in terrestrial ecosystems. *Global Change Biology*, 23, 3808-3824.

L100: "a high-resolution map" to "high-resolution maps"

Response: Thank you for these comments. We have corrected these inappropriate descriptions in the text and cited the recommended papers (L93-94).

L111-112: are all plots the same size for forests, shrublands, and grasslands?

Response: Thanks for the suggestion. We stated the plot sized in different vegetation types. Please see 2.1(L113-L114).

"At each site, one  $20 \times 50$  m2 plot was set for forests, three replicated  $5 \times 5$  m2 plots were set for shrublands, and ten  $1 \times 1$  m2 plots were established for grasslands."

L123-126: you may give references for the methods here. Equation 1: should the sum symbol with "i = 0" to "n" added? n is the total number of plant species. Similar for Equation 2.

Response: Thank you for the suggestion. We made this correction at L135 and L147 and corrected the description in text.

L135:

" $N(P) = \sum_{i=0}^{n} B_i \times \theta_i$

N(P) represents the community-level N or P density (Mg ha-1); n is the total number of plant species in one site..."

L147:

 $"SND(SPD) = \sum_{i=0}^{n} (1 - \delta_i) \times \rho_i \times C_i \times T_i / 10$

where *SND* (*SPD*) is the total N or P density of the soil within top 1 m (Mg ha-1); *n* is the total number of soil layers (ranging from one to five) in one site..."

L259, the unit of 5?

L269-281: one digit after decimal is enough and would be easier to read.

Response: Thank you for this comment. We made the corrections to the description of results according to your suggestions.

L295: I can't understand the reason. The reason may be expanded to be clear. Response: Thank you for the suggestion. We have changed the text in this part of the manuscript(L323-L324).

"Models for soil showed relatively poorer accuracy than models for plant organs and litter (Fig. 4 & 5), partly because that soil N and P were largely influenced by geological conditions, soil age and parent material (Gray and Murphy, 2002; Buol and Eswaran, 1999) (Doetterl et al., 2015), which were not included in our analysis because of the limited data availability. The can be evidenced by the decreasing validation R2 of the models for soil N densities and P densities and concentrations with soil depths (Fig. 5 and S3)."

L303: "the predicted SDs" is confusing. You may mean "SDs of the predictions" L313: remove "the"

L330: You may also cite the classic paper on this topic: Walker, T.W. & Syers, J.K. (1976). The fate of phosphorus during pedogenesis. *Geoderma*, 15, 1-19. L346: not necessarily more accurate predictions, depends on whether the models are informed by measurements such as those used in this study. "could" may be changed to "may".

Response: Thank you for the suggestions. We made these corrections and cited this paper at L370.

Fig. 3 color legend in panel (a) may include colors only for leaf/stem/root, with colors for vegetation/soil moved to panel (c), because panel (a) and (b) do not have vegetation vs. soil.

Response: Thank you for the suggestion. We moved the legend for vegetation/soil to panel (c). Please see Fig. 1.

Fig. 4: is there a reason for the slopes to be consistently higher than 1.0 across ecosystem components and nutrients? It seems to be a systematic bias in the models: overestimate when observed values are low and underestimate when observed values are high.

Response: Thank you for the suggestion. We changed the prediction method, and the slopes and intercepts are close to 1 and 0, respectively. Please see Fig 4 and 5.

**Tables and figures in the revised manuscript**

| 2 | Table.1. N and P | stocks of vegetation, | litter, soil and total | l ecosystem in forests, | shrublands and | grasslands in China. |
|---|------------------|-----------------------|------------------------|-------------------------|----------------|----------------------|
|---|------------------|-----------------------|------------------------|-------------------------|----------------|----------------------|

| Vegetation | Vegetation | Area          | N pool (Ta)  |        |            |           | D pool (Ta) |        |        |           |
|------------|------------|---------------|--------------|--------|------------|-----------|-------------|--------|--------|-----------|
| type group | type       | $(10^{6} ha)$ | N pool (1g)  |        |            |           | P pool (1g) |        |        |           |
|            |            |               | Vegetation   | Soil   | Litter     | Ecosystem | Vegetation  | Soil   | Litter | Ecosystem |
| Forest     | EBF        | 40.6          | 18.0         | 476.4  | 1.7        | 496.1     | 1.7         | 154.8  | 0.1    | 156.6     |
|            | DBF        | 66.3          | 43.1         | 811.3  | 3.7        | 858.1     | 6.9         | 346.5  | 0.4    | 353.8     |
|            | ENF        | 83.8          | 28.4         | 952.8  | 2.8        | 984.0     | 3.7         | 349.2  | 0.2    | 353.1     |
|            | DNF        | 11.5          | 5.6          | 177.7  | 0.5        | 183.8     | 1.5         | 73.6   | 0.1    | 75.2      |
|            | MF         | 9.6           | 4.6          | 107.6  | 0.5        | 112.8     | 0.9         | 41.5   | 0.1    | 42.4      |
|            | subtotal   | 211.9         | 99 .8 | 2525.8 | 9.3 | 2634.9    | 14.6        | 965.6  | 0.9    | 981.1     |
|            |            |               |              |        |            |           |             |        |        |           |
| Shrubland  | EBS        | 18.7          | 2.1          | 213.6  | 0.5        | 216.2     | 0.2         | 80.9   | < 0.1  | 81.1      |
|            | DBS        | 48.7          | 5.5          | 570.9  | 1.2        | 577.6     | 0.5         | 233.6  | 0.1    | 234.2     |
|            | ENS        | 1.0           | 0.1          | 12.4   | < 0.1      | 12.5      | < 0.1       | 4.9    | < 0.1  | 4.9       |
|            | SS         | 11.9          | 0.5          | 66.1   | 0.1        | 66.7      | < 0.1       | 61.6   | < 0.1  | 61.6      |
|            | subtotal   | 80.3          | 8.1          | 863.0  | 1.8        | 873.0     | 0.7         | 381.0  | 0.1    | 381.8     |
|            |            |               |              |        |            |           |             |        |        |           |
| Grassland  | ME         | 44.2          | 11.6         | 806.9  | 0.1        | 818.5     | 0.9         | 247.2  | < 0.1  | 248.0     |
|            | ST         | 137.4         | 21.3         | 1348.5 | 0.3        | 1370.1    | 1.5         | 573.1  | < 0.1  | 574.6     |
|            | TU         | 22.8          | 2.3          | 230.4  | 0.1        | 232.8     | 0.2         | 112.9  | < 0.1  | 113.2     |
|            | SG         | 103.8         | 13.6         | 860.6  | 0.1        | 874.4     | 0.9         | 506.3  | < 0.1  | 507.2     |
|            | subtotal   | 308.2         | 48.8         | 3246.4 | 0.6        | 3295.8    | 3.5         | 1439.5 | < 0.1  | 1443.0    |
| Total      |            | 600.4         | 156.7        | 6635.2 | 11.7       | 6793.1    | 18.8        | 2786.1 | 1.0    | 2806.0    |

- 3 EBF, evergreen broadleaf forest; DBF, deciduous broadleaf forest; ENF, evergreen needle-leaf forest; DNF, deciduous needle-
- 4 leaf forest; MF, broadleaf and needle-leaf forest; EBS, evergreen broadleaf shrub; DBS, deciduous broadleaf shrub; ENS,
- 5 evergreen needle-leaf shrub; SS, sparse shrub; ME, meadow; ST, steppe; TU, tussock; and SG, sparse grassland.

| Vegetation type group | Vegetation type | Area (10 6 ha) | N pool (Tg) |       | P pool (Tg) |       |       |       |
|-----------------------|-----------------|---------------------------|-------------|-------|-------------|-------|-------|-------|
|                       |                 |                           | Leaf        | Stem  | Root        | Leaf  | Stem  | Root  |
| Forest                | EBF             | 40.6                      | 3.9         | 10.1  | 4.0         | 0.3   | 1.0   | 0.3   |
|                       | DBF             | 66.3                      | 6.1         | 26.6  | 10.5        | 0.6   | 4.6   | 1.6   |
|                       | ENF             | 83.8                      | 8.6         | 13.4  | 6.4         | 0.9   | 2.0   | 0.8   |
|                       | DNF             | 11.5                      | 1.3         | 2.9   | 1.4         | 0.2   | 0.9   | 0.3   |
|                       | MF              | 9.6                       | 1.0         | 2.6   | 1.0         | 0.1   | 0.7   | 0.2   |
|                       | subtotal        | 211.9                     | 21.0        | 55.5  | 23.4        | 2.1   | 9.2   | 3.3   |
| Shrubland             | EBS             | 18.7                      | 0.6         | 0.7   | 0.7         |

9 Fig. 1. Frequency distributions of N densities in soil, roots, leaves, litter and woody stems in

10 forests (a-e), shrublands (f-j) and grasslands (k-n) in China.

---

## Author Response (AR2)

8 Sept, 2021
Dr. David Carlson
Senior Chief Editor
*Earth System Science Data*

**RE: ESSD-2020-398**

Dear Dr. Carlson,

Thank you very much for handling our manuscript entitled *Patterns of nitrogen and phosphorus pools in terrestrial ecosystems in China* (ESSD-2020-398). We would like to resubmit the manuscript after consideration and revision following your comments.

You pointed out the problems in the derived data from this research and the access to the datasets for analysis. As to the former, we had uploaded a series of *.tif files and one README file during the previous revision stage to present our predictions. Each *.tif file contains nitrogen or phosphorus information of one certain ecosystem component and can be read and manipulated with GIS software such as ArcGIS or package *raster* in R, with which the underlying nutrient data can be extracted. One recent paper published on ESSD (https://doi.org/10.5194/essd-13-3927-2021) also uploaded similar *.tif files. Therefore, we think that our files could be operable as well. However, we are happy to upload other formats such as *.csv, if you insist (which can be converted from those *.tif files using GIS software.).

In terms of the access to the datasets for analysis, we have uploaded site coordinates and *.tif layers of environmental factors including MAT, MAP, elevation, EVI and vegetation types to the data repository. We hope that the field survey data of nutrients could be exempted for now if possible for the following reasons. First, the main idea of this research is just to generate spatial data products on nitrogen and phosphorus, while the original nutrient dataset contains information beyond this topic. Second, the field data was collected and shared by many principle investigators, and our as well as other PIs' ongoing studies can encounter trouble if the original nutrient dataset is published now. However, if you insist that it is necessary to publish the original dataset, we agree to upload it after communication with other groups.

Specifically, we made the following revisions. First, we divided all *.tif layers of nutrients in to three categories (density, concentration and ratio), numbered them with corresponding figure numbers in the manuscript, and re-upload them to the data repository. Second, the geographic coordinates of field sites and *.tif layers of predictors were uploaded as well, which was mentioned in the section of Data accessibility as "…are available from the Dryad Digital Repository along with the geographic coordinates of field sites and layer files of environmental factors for prediction" (line 219). We also revised the links of GTOPO30 (line 161) and MODIS EVI data (line 165). Additionally, we made some minor corrections on texts.

We hope you will find our revision satisfactory. Please contact us if you have any questions. We look forward to hearing from you.

With best regards,

Sincerely yours,

Dr. Zhiyao Tang (on behalf of the author team)
Department of Ecology
Peking University
Beijing, China

---

## Author Response (AR3)

23 Sept, 2021
Dr. David Carlson
Senior Chief Editor
*Earth System Science Data*

**RE: ESSD-2020-398**

Dear Dr. Carlson,

Thank you very much for pointing out this problem. We have added the corresponding labels to the panels in Fig 1 and 2 following your comments. Please see the updated manuscript for details.

We hope you will find our revision satisfactory. Please contact us if you have any questions. We look forward to hearing from you.

With best regards,

Sincerely yours,

Dr. Zhiyao Tang (on behalf of the author team)
Department of Ecology
Peking University
Beijing, China